# Shape Sensing of Cantilever Column Using Hybrid Frenet–Serret Homogeneous Transformation Matrix Method

**DOI:** 10.3390/s24082533

**Published:** 2024-04-15

**Authors:** Peng Zhang, Duanshu Li, Ran An, Patil Devendra

**Affiliations:** 1Department of Civil Engineering, Dalian Maritime University, Dalian 116026, China; peng.zhang47@dlmu.edu.cn (P.Z.); ldsmdn1110@dlmu.edu.cn (D.L.); anran_0103@dlmu.edu.cn (R.A.); 2Department of Mechanical Engineering, BITS Pilani K K Birla Goa Campus, Zuarinagar 403726, India

**Keywords:** shape sensing, displacement reconstruction, Frenet–Serret framework, homogeneous transmission matrix

## Abstract

The Frenet–Serret (FS) framework stands as a pivotal tool in shape sensing for various infrastructures. However, this tool suffers from accumulative errors, particularly at inflection points where the normal vector undergoes sign changes. To minimize the error, the traditional FS framework is modified by incorporating the homogeneous matrix transformation (HMT) method for segments containing inflection points. Additionally, inclination information is also used to calculate the unit tangent vector and the unit norm vector at the start point of each segment. This novel approach, termed the FS-HMT method, aims to enhance accuracy. To validate the effectiveness of the proposed method, a simulation of a cantilever column was conducted using finite element software ANSYS 19.2. The numerical results demonstrate the capability of the proposed method to accurately predict curves with inflection points, yielding a maximum error of 1.1%. Subsequently, experimental verification was performed using a 1 m long spring steel sheet, showcasing an error of 4.9%, which is notably lower than that of the traditional FS framework. Our proposed modified FS framework exhibits improved accuracy, especially in scenarios involving inflection points. These findings underscore its potential as a valuable tool for enhanced shape sensing in practical applications.

## 1. Introduction

Recent decades have witnessed fast development in the field of structural health monitoring (SHM) techniques for a vast set of applications to meet global needs. Currently, the mainstream structural health monitoring methods are based on point sensors, which analyze various characteristics of structures through the real-time online monitoring of structural physical quantities, such as strain, displacement, inclination, force, and acceleration, at key nodes [1,2,3,4,5]. Simultaneously, shape-sensing (SS) techniques are also developed to estimate the condition of the structural parts where no sensors are installed.

SS techniques aim to reconstruct the deformed shape of a structure subjected to load or actions. Currently, several shape-sensing methods or algorithms have been proposed by various research teams from different backgrounds for different purposes: One major group of shape-sensing methods is based on image processing or computer vision technologies. In one such technique—high-precision CCD (charge-coupled device)—cameras are used to capture images, and data processing is carried out through trained neural networks such as recurrent neural networks (RNNs) [6,7], integral imaging [8], or convolutional neural networks (CNNs) [9]. However, the accuracy of this group of shape-sensing technologies is dependent on the image quality. Moreover, the image-based technology can only be applied to the visible part of a structure. The shapes of the invisible components (e.g., the pile foundation) have to be reconstructed using other algorithms, such as Ko’s displacement theories [10], the inverse finite element method [11,12,13,14], modal-based algorithms [15,16], and the Frenet–Serret framework [17,18,19,20,21,22,23,24]. 

The Frenet–Serret (FS) formulas, which were initially used to describe the kinematic properties of a particle moving along a differentiable curve in Euclidean space, have been employed to reconstruct the shape of a curve. In 2012, Moore and Rogge [25] proposed to convert strain to the curvature and torsion of spatial curves. Concurrently, they pioneered the integration of the FS framework into spatial shape reconstruction, marking the inaugural application of this methodology. Subsequently, leveraging the FS framework, Wu [26] addressed the challenge of compensating for the impact of torsion or temperature on algorithmic precision by integrating double Fiber Bragg Grating (FBG) cross-laying with the FS framework’s design. In a parallel effort, Al-Ahmad [27] introduced a two-step calibration method within the FS framework to enhance accuracy, specifically focusing on torsion compensation. Meanwhile, He [28] developed a temperature compensation algorithm to enhance measurement accuracy. This algorithm was seamlessly incorporated into the FS framework for shape reconstruction. However, none of those approaches addresses the issue of the FS framework when the normal vectors abruptly change signs at inflection points. To effectively mitigate measurement errors and cumulative inaccuracies, Tian [29] implemented a rotation-minimizing frame (RMF) to solve the FS frame for shape restoration. This approach successfully reconstructed structural curves even under complex deformation states. Additionally, Tian segmented the structure into multiple sections to minimize cumulative errors. In a parallel effort, Wang [30] performed sensor calibration to reduce measurement errors and subsequently restored the structural shape using the FS framework. Zhu [31], on the other hand, employed a photoelectric sensor to reconstruct the overall shape of a flexible arm based on the FS framework. Furthermore, Zhu devised a specific calibration procedure tailored for the shape reconstruction/sensing of flexible arms. Based on the FS framework and error transfer theory, Liu [32] established a comprehensive model to describe the relation between Fiber Bragg Grating (FBG) calibration direction, placement angle, and shape restoration errors. The effectiveness of this method in calibrating curvature and ensuring shape restoration accuracy was verified through simulations. However, it is worth noting that this approach, grounded in the FS framework, still faces challenges in dealing with significant reduction errors in curves featuring inflection points.

In summary, the current challenges in shape restoration algorithms encompass the following issues:The shape restoration algorithm based on the FS framework exhibits a tendency to abruptly alter the normal vector at inflection points, particularly when the bending direction of the curve changes. This behavior leads to significant reconstruction errors after such inflection points.The shape reconstruction algorithm, grounded in the FS framework, relies on knowledge of the position information of the preceding point. Subsequently, utilizing data collected by sensors, it recursively determines the coordinates of subsequent points to achieve the overall structure’s curve reconstruction. However, this approach introduces error accumulation, resulting in a substantial error towards the end of the reconstruction process.

To address these challenges, this paper proposes a modified shape reconstruction algorithm that integrates strain data with inclination data, leveraging the FS framework and the homogenous transmission matrix. This novel approach aims to rectify distortions at inflection points and mitigate the issue of cumulative errors encountered by the current algorithm. The remainder of this paper is organized as follows: Section 2 introduces the theory of the FS framework. Section 3 explains the modified shape reconstruction algorithm. In Section 4 and Section 5, the proposed algorithm is verified through numerical and experimental investigations. Finally, Section 6 summarizes the findings and limitations of this study.

## 2. Frenet–Serret Framework-Based Shape-Sensing Algorithm

The FS formulas describe the shape of a three-dimensional curve using the unit vector tangent to the curve, the normal unit vector, and the binormal unit vector of the curve. When applied to shape reconstruction of a structure, the following three steps are conducted (as shown in Figure 1):Install strain sensors to measure the strain at several points of the structure and calculate the curvature;Obtain the bending curvature function using the cubic spline interpolation method;Substitute the curvature into the FS framework to obtain the set of differential equations, and then solve the differential equations to obtain the position information of each point.

In differential geometry, the FS formulas describe the motion of a point along a continuous differentiable curve (as shown in Figure 2). The shape of the curve can be determined by substituting the curvatures into the FS frame.

Assume that the shape of the curve to be restored can be expressed as
(1)$$r(s)=x(s)\vec{i}+y(s)\vec{j}$$
where s is the arc length coordinate, and $\vec{i}$ and $\vec{j}$ are the unit vectors in the *x*-axis and *y*-axis directions, respectively.

Define T(s) and N(s) as the unit tangent vector and the unit normal vector of the curve, respectively. They are represented as
(2)$$T(s)=\frac{dr(s)}{ds}=\frac{dx(s)}{ds}\vec{i}+\frac{dy(s)}{ds}\vec{j}$$
(3)$$N(s)=\frac{\frac{dT(s)}{ds}}{\left\|\frac{dT(s)}{ds}\right\|}=\frac{\frac{d^{2}x(s)}{ds^{2}}\vec{i}+\frac{d^{2}y(s)}{ds^{2}}\vec{j}}{\sqrt{{\left(\frac{d^{2}x(s)}{ds^{2}}\right)}^{2}+{\left(\frac{d^{2}y(s)}{ds^{2}}\right)}^{2}}}$$

The derivatives of T(s) and N(s) can be calculated as
(4)$$\begin{array}{l} \frac{dT(s)}{ds}=k(s)N(s)\\ \frac{dN(s)}{ds}=-k(s)T(s) \end{array}$$

This system of equations can be solved using the fourth-order Runge–Kutta method, assuming initial conditions of r(0)=(0,0), k(0)=0, T(0)=(1,0), and N(0)=(0,1). And finally, the shape the curve r(s) can be calculated as
(5)r(s)=∫T(s)ds+r(0)

However, the FS framework has inherent defects and discontinuities at the inflection points, resulting in accumulative errors and the failure of shape reconstruction. Therefore, a new algorithm is of necessity for shape sensing for curves with inflection points.

## 3. Modified FS Framework Based on Strain and Inclination Data

In order to reconstruct the shape of a curve with inflection points, this study introduces the inclination sensor into the strain-based FS framework. Those inclination sensors serve two purposes: first, to divide the curve into several segments, and second, to approximate the curve’s second-order derivative, which can indicate if an inflection point occurs within any segment. If there is no inflection point in a segment, the traditional FS framework can be applied to reconstruct the geometry of the segment. Otherwise, a homogeneous transformation matrix shall be utilized to obtain the shape of the segment. The specific steps are listed below and a flowchart is also provided in Figure 3 to illustrate the process:Define an initial condition of *r*(0) = (0,0), *T*(0) = (1,0), and *N*(0) = (0,1).Divide the curve into several segments using the strain gauges.Measure the strain at each point with a strain gauge, and then calculate the curvature function using Equation (5). This step uses the same interpolation method as presented in Section 2.Determine if an inflection point exists within the segment. Then, perform shape sensing on the segment using either the traditional FS framework or the homogeneous transformation matrix.Calculate the *T* and *N* of the start point of each segment using inclination data and then obtain the whole shape of the curve.

### 3.1. Shape Sensing with Homogenous Transformation Matrix

In the proposed algorithm, the shape of a segment will be reconstructed using either the traditional FS framework or the homogeneous transformation matrix (HTM). If the strains at the start and the end points of the segment have the same sign, no inflection point exists within the segment. On the other hand, if the strains have different signs, an inflection point occurs and the traditional FS framework shall be replaced by the homogeneous transformation matrix.

Figure 4 illustrates the schematics of the homogeneous transformation matrix. Establish a moving coordinate system $M_{i}(M_{ix},M_{iy},M_{iz})$ at each point ($O_{1},O_{2},O_{3},\cdots ,O_{n}$) and an absolute coordinate system *F* at $O_{0}$. The coordinate of point $O_{i+1}$ in the moving coordinate system $M_{i}(M_{ix},M_{iy},M_{iz})$ is denoted as $\left(a_{i+1},b_{i+1},c_{i+1}\right)$. The coordinate of point $O_{i+1}$ in the absolute coordinate system is denoted as $\left(x_{i+1},y_{i+1},z_{i+1}\right)$. $\partial_{i}$ is the angle between the curve and the *x*-axis of the moving coordinate system $M_{i}$. The arc length between the two points is *s*. Then, the coordinates of the *i* + 1 point in the moving coordinate system $M_{i}$ can be expressed as
(6)$$\begin{array}{l} a_{i+1}=\rho_{i+1}(1-\cos\theta_{i})\cos\partial_{i}\\ b_{i+1}=\rho_{i+1}(1-\cos\theta_{i})\sin\partial_{i}\\ c_{i+1}=\rho_{i+1}\sin\theta_{i} \end{array}$$
where $\rho_{i+1}$ is the radius of the curve $\overset{⌢}{O_{i}O_{i+1}}$, while $\theta_{i}$ is the corresponding angle. Each moving coordinate system $M_{i}$ can be converted to the absolute coordinate system *F* through the transformation matrix $T_{i}$:(7)$$F=T_{i}M_{i}$$

The coordinate of point $O_{i+1}$ in the absolute coordinate system is then determined by
(8)$$\left\{\begin{array}{l} x_{i+1}\\ y_{i+1}\\ z_{i+1} \end{array}\right\}=T_{i}\left\{\begin{array}{l} a_{i+1}\\ b_{i+1}\\ c_{i+1} \end{array}\right\}$$

Note that each moving coordinate system can be obtained by the previous moving coordinate system:(9)$$M_{i+1}=t_{i+1}M_{i}$$
where
(10)$$t_{i+1}=PR_{1}R_{2}R_{3}$$

Here, *P* is a translation matrix, and $R_{1},R_{2},R_{3}$ are three rotation matrices. Thus, the recursive relationship between the transformation matrices can be obtained:(11)$$T_{i+1}=T_{i}t_{i+1}^{-1}$$

In Equation (10), transformation matrices $R_{1i},R_{2i},R_{3i}$ and $P_{i}$ are determined as follows:(12)$$R_{i1}=\left(\begin{array}{cccc} \cos(\partial_{i}) & -\sin(\partial_{i}) & 0 & 0\\ \sin(\partial_{i}) & \cos(\partial_{i}) & 0 & 0\\ 0 & 0 & 1 & 0\\ 0 & 0 & 0 & 1 \end{array}\right)$$
(13)$$R_{i2}=\left(\begin{array}{cccc} \cos(\theta_{i}) & 0 & -\sin(\theta_{i}) & 0\\ 0 & 1 & 0 & 0\\ \sin(\theta_{i}) & 0 & \cos(-\theta_{i}) & 0\\ 0 & 0 & 0 & 1 \end{array}\right)$$
(14)$$R_{i3}=\left(\begin{array}{cccc} \cos(-\partial_{i+1}) & -\sin(-\partial_{i+1}) & 0 & 0\\ \sin(-\partial_{i+1}) & \cos(-\partial_{i+1}) & 0 & 0\\ 0 & 0 & 1 & 0\\ 0 & 0 & 0 & 1 \end{array}\right)$$
(15)$$P_{i}=\left(\begin{array}{cccc} 1 & 0 & 0 & a_{i+1}\\ 0 & 1 & 0 & b_{i+1}\\ 0 & 0 & 1 & c_{i+1}\\ 0 & 0 & 0 & 1 \end{array}\right)$$

### 3.2. Calculate T, N Using Inclination 

In the proposed new framework, if any segment of the curve has an inflection point, its shape will be reconstructed using the homogenous transformation matrix instead of the traditional FS framework. However, to integrate the shape of this segment into the whole curve, the unit tangent vector *T* and the unit normal vector *N* at the start point of the segment shall be calculated. In this study, they are calculated using the observed inclination *q* at the start point:(16)$$T=\left(T_{x},T_{y}\right)=(\cos q,\sin q)$$
(17)$$N=\left(N_{x},N_{y}\right)=(\sin q,-\cos q)$$

## 4. Numerical Verification

In order to verify the proposed algorithm’s performance in reconstructing the shape of a curve, a cantilever column is simulated in ANSYS 19.2. The structure is a 1 m × 0.2 m spring steel sheet with its bottom fixed to the ground. The thickness is 1 mm and the elastic modulus is $2.1\times 10^{11}$ Pa. The numerical model of this structure is studied in ANSYS. A sinusoidal load is applied to excite the third-order vibration mode.

The strain and inclination are substituted into the traditional and modified FS framework, and the shape of the structure is calculated and compared in Figure 5. It can be seen that both algorithms could accurately restore the curve before the inflection point. After the inflection point, the proposed algorithm could also reconstruct the shape, with a maximum relative error of 1.1% at the tip of the structure. However, the traditional FS framework failed due to normal vector changing signs at the inflection point. The maximum relative error reached 69.28% with the traditional FS. 

## 5. Experimental Verification

### 5.1. Experimental Design

The experimental system mainly comprises an optical platform, a spring steel sheet, triangle-shaped connectors, strain sensors, inclinometers, displacement sensors, a data acquisition instrument, and a computer, as shown in Figure 6. The optical platform serves as the foundation of the cantilever column, which is a 1m high steel sheet with a width of 200 mm. The steel sheet was fixed to the optical platform via four triangle-shaped connectors. Two screws were also fixed to the triangle-shaped connectors to apply lateral displacement to the cantilever column. The inclinometer used in the experiment has a length of 57 mm, a width of 57 mm, a thickness of 39 mm, a mass of 93 g, a resolution of 0.05°, and an accuracy of ±0.01°.

A total of ten strain sensors and four inclinometers were arranged along the height of the steel sheet as shown in Figure 6b. The sampling rate was set to 60 Hz. After collecting the data, the position information, strain information, and inclination information of each point are obtained, input into the computer, substituted into the algorithm for calculation, and the curve of the structure is reconstructed.

### 5.2. Data Processing

As mentioned previously, the FS framework and the homogeneous transformation matrix can describe the geometry of a curve based on the curvature data. To restore the curve of the steel sheet, the strain measured at each point should be converted into a curve. As shown in Figure 7a, the original length of the steel sheet is L and the distance between the surface and the neutral axis is *d*. When the structure is bent by the radius *R* along the neutral axis, the upper (outer) surface is stretched, resulting in tensile strain, ε, and increased length, $L_{1}$. Meanwhile, the lower (inner) surface is compressed, resulting in compressive strain, −ε, and a compressed length, $L_{2}$, as shown in Figure 7a. The bent shape is shown in Figure 7b.

According to the plane section assumption, the relation of those geometric parameters can be expressed as
(18)$$L_{1}=(1+\varepsilon )L$$
(19)$$L_{2}=(1-\varepsilon )L$$
(20)$$\frac{L}{R}=\frac{L_{1}}{R+d}=\frac{L_{2}}{R+d}$$

Substituting Equations (18) and (19) into Equation (20) yields:(21)$$\varepsilon =\frac{d}{R}$$
or
(22)ε=kd

Here, *k* is the curvature of the curve, and *R* is the radius of the curvature. 

According to the measured data of the strain sensor and the calculation mentioned above, the curvature is calculated and is shown in Figure 8.

Based on Equation (22), the bending curvatures can be estimated using the strain signals observed at discrete distributed nodes of the curve. A cubic spline interpolation process shall be performed to obtain the curvature function of the curve.

Assume that the bending curvature can be expressed by a cubic spline function as
(23)$$k(s)=a_{i}+b_{i}\times (s-s_{i})+c_{i}\times {\left(s-s_{i}\right)}^{2}+d_{i}\times {\left(s-s_{i}\right)}^{3}$$
where s denotes the arc length coordinates and $a_{i}$, $b_{i}$, $c_{i}$, $d_{i}$ are the four coefficients to be determined using the interpolation continuity principle and the first-order and second-order differential continuity principle at the *i*th node:(24)$$\{\begin{array}{l} k_{i}(s_{i})=c_{i}\\ k_{i}(s_{i+1})=c_{i+1}\\ {k^{\prime }}_{i}(s_{i+1})={k^{\prime }}_{i+1}(s_{i+1})\\ {k^{\prime \prime }}_{i}(s_{i+1})={k^{\prime \prime }}_{i+1}(s_{i+1}) \end{array}$$

### 5.3. Experimental Results

Two experimental cases were performed to compare the accuracy of the proposed FS-HMT and the traditional FS framework. In the first experimental case, Screw 1 was adjusted to different positions to impose displacements on the tip of the steel sheet to 0.2 m, 0.3 m, and 0.5 m, successively. No inflection point occurred in this experimental case. The reconstructed shape and error of the two algorithms are compared in Figure 9. It can be observed that both the traditional FS framework and the FS-HMT method demonstrated satisfying accuracy. The maximum error of the traditional FS framework observed is 2.98%, while that of the FS-HMT method is 2.77%. In general, the FS-HMT method is more accurate than the traditional FS framework. This can be attributed to the fact that the proposed FS-HMT method utilizes observed inclination information to calculate *T* and *N* at the start point of each segment of the curve, consequently reducing the accumulative error. 

In the second experimental case, Screw 1 was still used to bend the steel sheet toward the left. Additionally, Screw 2 was adjusted to different positions to bend the upper part of the steel sheet toward the right. Consequently, an inflection point was generated between Screw 1 and Screw 2. Figure 10 compares the reconstructed shape and errors of the two algorithms. 

It can be seen from Figure 10 that both the traditional FS framework and the proposed FS-HMT method can accurately reconstruct the shape of the cantilever column between the bottom and the inflection point. However, the traditional FS framework fails to detect the inflection point, and the shape estimated by the traditional FS framework still bends toward the left after the inflection point. Due to this, considerable accumulative error can be detected in the traditional FS framework. On the other hand, the proposed FS-HMT method can detect the inflection point and utilize the homogenous transformation matrix to reconstruct the shape of the segment with an inflection point. The proposed method also successfully constructed the shape post inflection point with a maximum error of only 4.9%. 

## 6. Conclusions

In this paper, a new shape-sensing algorithm, i.e., the FS-HMT method, was proposed by combining the traditional FS framework and the homogenous transformation matrix. Numerical and experimental studies have been conducted to verify the accuracy of the proposed method. The main conclusions of this study are presented as follows:Compared with the traditional FS framework, the proposed FS-HMT method utilizes the homogenous transformation matrix to reconstruct the shape of the segment with inflection points and utilizes inclination information to calculate the *T* and *N* of the starting point of a segment.Regarding the curves without an inflection point, both the FS-HMT and the traditional FS framework can accurately reconstruct the deformed shape of the structure. The FS-HMT method slightly improved the accuracy compared with the traditional FS framework.Regarding the curves with an inflection point, the traditional FS framework failed due to accumulative error after the inflection point, while the FS-HMT framework demonstrated satisfying accuracy, with a maximum error of only 4.9%.

Though the proposed FS-HMT method has demonstrated satisfying accuracy to reconstruct the shape of a curve with inflection points, the optimization of the sensors (number and placement) was not considered in this study. In future, the influence of geometric shape and size shall also be investigated to further improve the accuracy and efficiency of this method. A comparison between FS-HMT and other shape-sensing technologies (e.g., inverse finite element method, coordinate transformation method, the parallel frame, and the rotation-minimizing frame) shall be performed to further reveal the performance of each method.

## Figures and Tables

**Figure 1 sensors-24-02533-f001:**
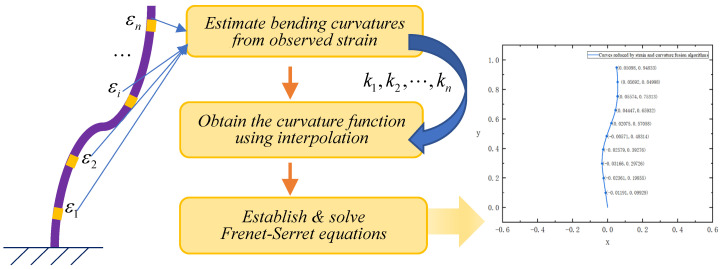
Process of the shape reconstruction algorithm based on the FS framework.

**Figure 2 sensors-24-02533-f002:**
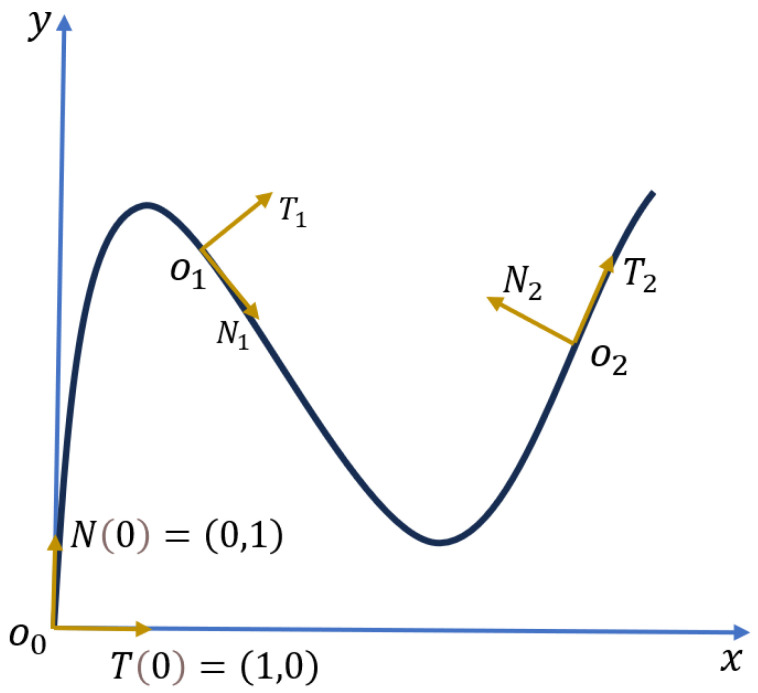
Schematics of Frenet–Serret framework.

**Figure 3 sensors-24-02533-f003:**
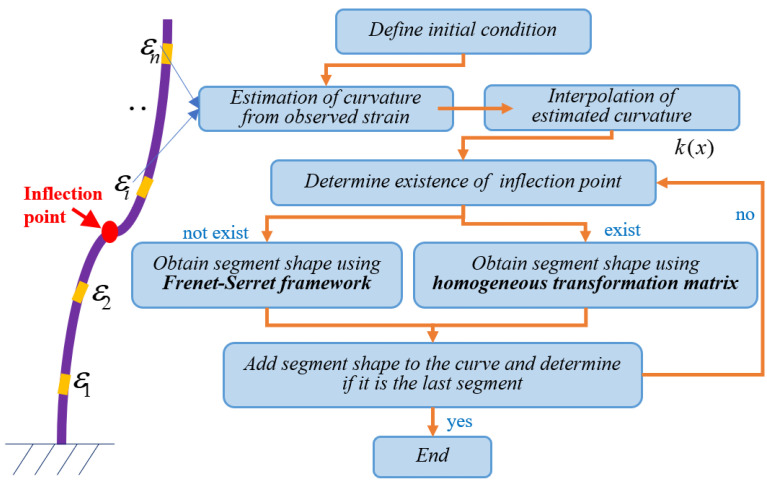
Process of shape reconstruction algorithm based on the proposed algorithm.

**Figure 4 sensors-24-02533-f004:**
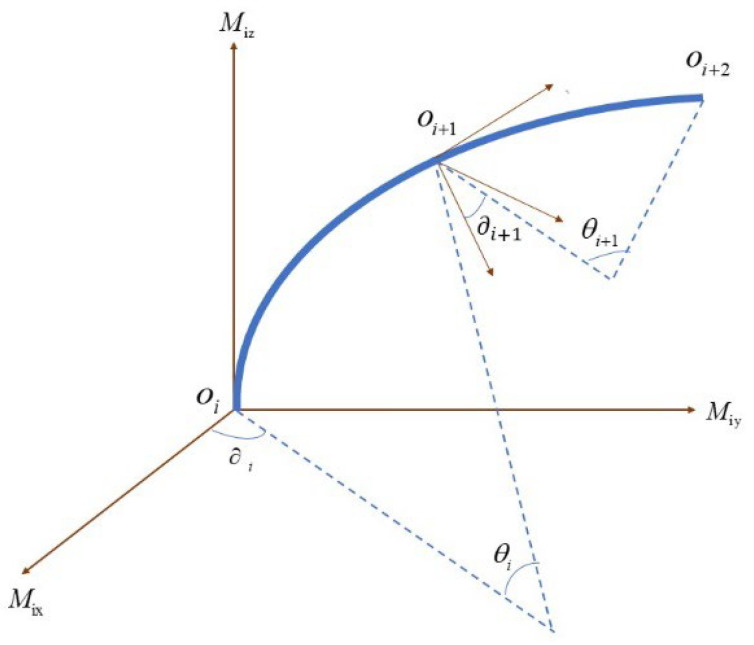
Shape sensing using a homogenous transmission matrix.

**Figure 5 sensors-24-02533-f005:**
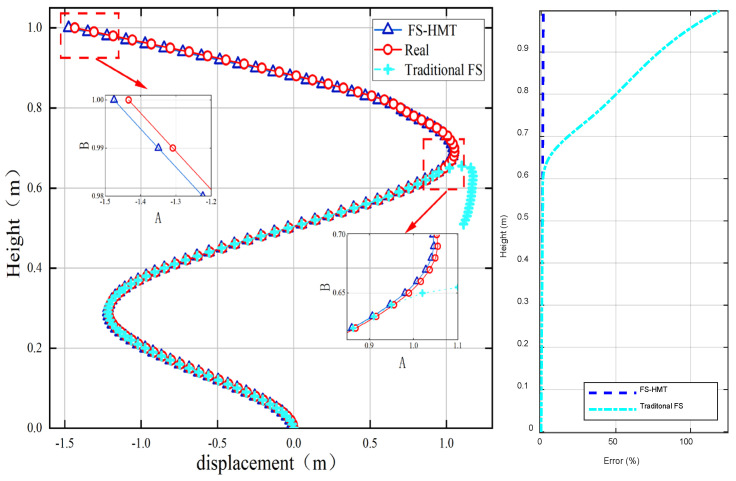
Comparison of the traditional FS framework and modified FS framework.

**Figure 6 sensors-24-02533-f006:**
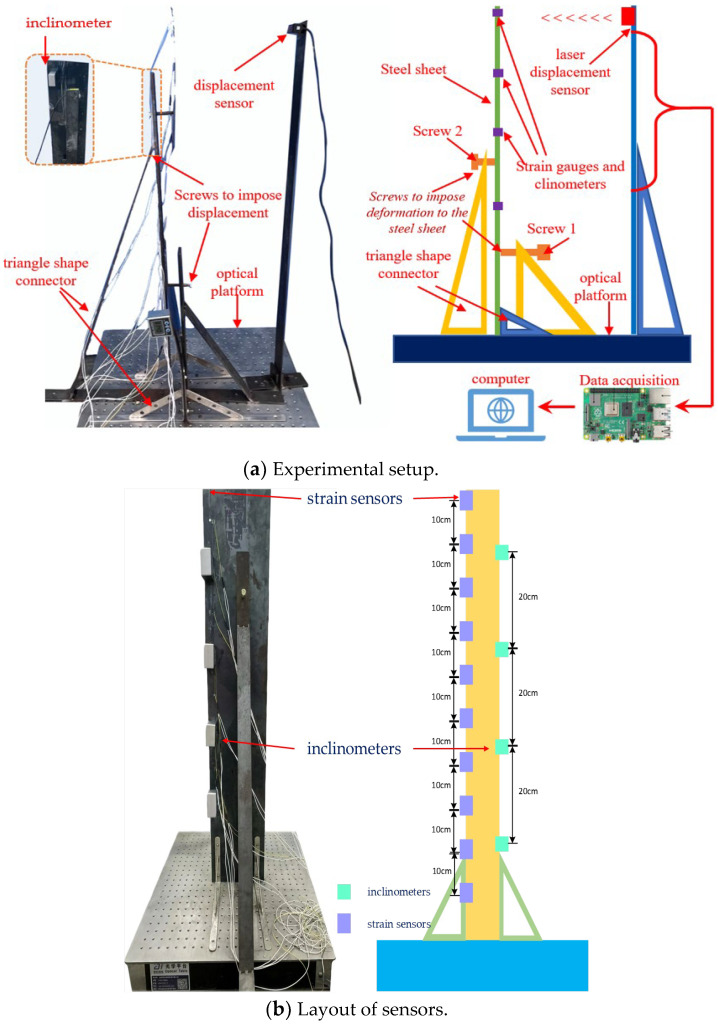
Experimental setup schematics.

**Figure 7 sensors-24-02533-f007:**
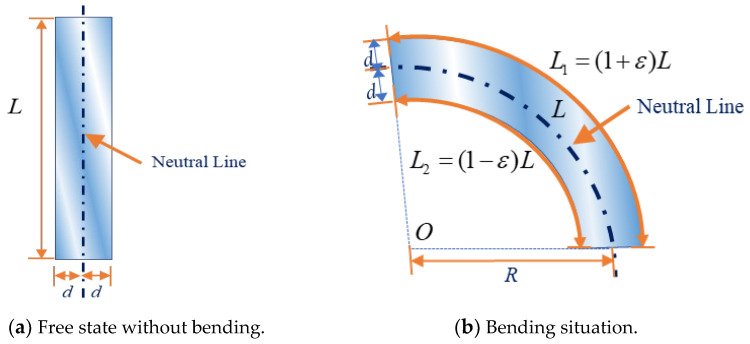
Steel sheet deformation subjected to bending.

**Figure 8 sensors-24-02533-f008:**
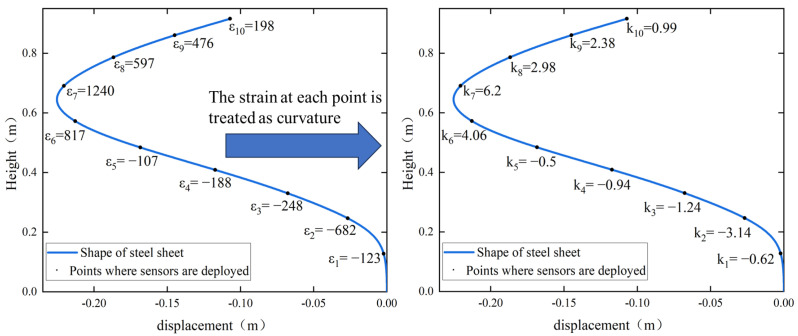
Post-processing representation of strain and curvature.

**Figure 9 sensors-24-02533-f009:**
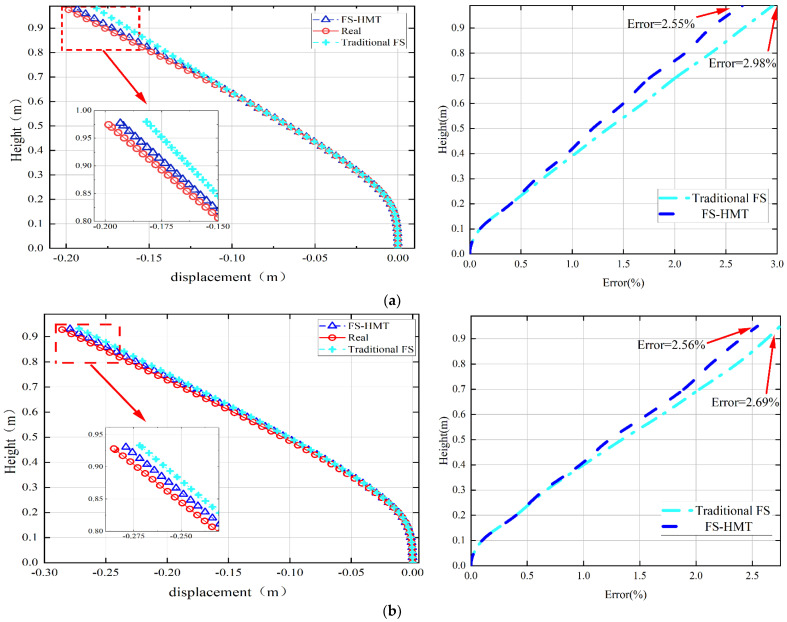
The reconstructed shape of the cantilever column using the FS-HMT method and traditional FS framework: (**a**) tip displacement of 0.2 m; (**b**) tip displacement of 0.3 m; (**c**) tip displacement of 0.5 m.

**Figure 10 sensors-24-02533-f010:**
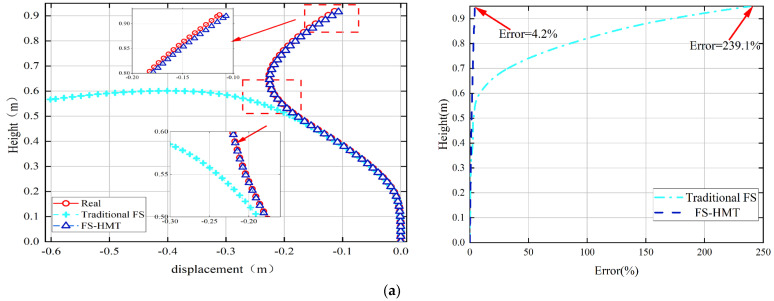
The reconstructed shape of the cantilever column using the FS-HMT method and traditional FS framework: (**a**) tip displacement of 0.091 m; (**b**) tip displacement of 0.089 m; (**c**) tip displacement of 0.087 m.

## Data Availability

The data used to support the findings of this study are available from the corresponding author upon request.

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
