# Peer review of "Shape Sensing of Cantilever Column Using Hybrid Frenet–Serret Homogeneous Transformation Matrix Method"

_sensors, 2024, doi:10.3390/s24082533_

Round 1

Reviewer 1 Report

Comments and Suggestions for Authors

First of all, I would like to emphasise the quality of the presentation of the paper and results. However, I have some questions and remarks

1. Please provide deciphering of the abbreviations in the introduction

2. Did you compare the efficiency of the method relative to the geometry of the elements?

3. Could you compare computational time of the proposed method with well-known?

Author Response

We would like to thank the editorial board and the anonymous reviewers of the manuscript for their careful reading and constructive feedback which has helped us to improve the manuscript. Following the given directives, we have prepared a revised version of the manuscript, where all of the comments and suggestions by reviewers have been addressed. Our response to Reviewer 1 is enclosed here.

Reviewer 2 Report

Comments and Suggestions for Authors

This manuscript introduces an enhanced Frenet-Serret framework incorporating the homogeneous matrix transformation (FS-HMT) for shape reconstruction. Validation of the proposed method was conducted through numerical and experimental campaigns, demonstrating its enhanced performance, particularly in scenarios involving inflection points. This contribution significantly advances the literature in this field and merits publication. To improve clarity, I have included some corrections and comments for your consideration.

1.      Fig. 6: Consider adding a zoom-in plot at the inflection point to demonstrate the performance of the proposed method more effectively.

2.      Line 259-260: It would be helpful to clarify the type and positioning of the sensors mentioned. For instance, are these sensors strain gauges? If so, how many strain gauges are positioned along the height of the steel sheet, and what is their configuration? Additionally, how many inclinometers are used, and what are their specifications? Fig. 7 only depicts four sensors, including the strain gauge and inclinometer, which might cause confusion.

3.      Figure 9: It is suggested to include subtitles for each figure, similar to Figure 8.

4.      Could the authors provide insight into the practical application requirements for inclinometers?

5.      Out-of-plane shape: What steps are necessary to achieve a comprehensive 3D shape reconstruction?

Author Response

We would like to thank the editorial board and the anonymous reviewers of the manuscript for their careful reading and constructive feedback which has helped us to improve the manuscript. Following the given directives, we have prepared a revised version of the manuscript, where all of the comments and suggestions by reviewers have been addressed. Our response to Reviewer 2 is enclosed here.

Reviewer 3 Report

Comments and Suggestions for Authors

1.The part 2 "Frenet-Serret framework-based shape sensing algorithm" seems unnecessary. Those formulation should be combined with the specific object of study.

2.How factors such as the size and geometry of the measured target will affect the results should be studied in depth.

3.It seems that the results should be compared with more methods (especially different sensor-based measurement methods) to better reflect the performance of the proposed algorithm. More rigorous comparative tests are recommended.

Author Response

We would like to thank the editorial board and the anonymous reviewers of the manuscript for their careful reading and constructive feedback which has helped us to improve the manuscript. Following the given directives, we have prepared a revised version of the manuscript, where all of the comments and suggestions by reviewers have been addressed. Our response to Reviewer 3 is enclosed here.

Round 2

Reviewer 3 Report

Comments and Suggestions for Authors

The manuscript provides reasonable answers and supplements to the questions raised.